# Efficacy and Safety of Melatonin Treatment in Children with Autism Spectrum Disorder and Attention-Deficit/Hyperactivity Disorder—A Review of the Literature

**DOI:** 10.3390/brainsci10040219

**Published:** 2020-04-07

**Authors:** Beata Rzepka-Migut, Justyna Paprocka

**Affiliations:** 1Department of Pediatric Neurology and Pediatrics, St. Queen Jadwiga’s Regional Clinical Hospital No 2, 35-301 Rzeszów, Poland; beata-rzepka@o2.pl; 2Department of Pediatric Neurology, Faculty of Medical Science in Katowice, Medical University of Silesia, 40-752 Katowice, Poland

**Keywords:** melatonin, autism spectrum disorder (ASD) attention deficit hyperactivity disorder (ADHD), sleep disorders

## Abstract

Autism spectrum disorder (ASD) and attention-deficit/hyperactivity disorder (ADHD) are neurodevelopmental disorders with disturbed melatonin secretion profile and sleep problems. The growing incidence of ASD and ADHD inspires scientists to research the underlying causes of these conditions. The authors focused on two fundamental aspects, the first one being the presentation of the role of melatonin in ASD and ADHD and the second of the influence of melatonin treatment on sleep disorders. The authors present the use of melatonin both in the context of causal and symptomatic treatment and discuss melatonin supplementation: Dosage patterns, effectiveness, and safety. Sleep disorders may have a different clinical picture, so the assessment of exogenous melatonin efficacy should also refer to a specific group of symptoms. The review draws attention to the wide range of doses of melatonin used in supplementation and the need to introduce unified standards especially in the group of pediatric patients.

## 1. Introduction

The circadian rhythm allows organisms to adapt to the changing environmental conditions resulting from the day and night cycle. The organic chemical compound N-acetyl-5-methoxytryptamine called melatonin is the main regulator of circadian rhythm and was first isolated in 1958. 62 years of intensive research provided us with information on the metabolism and function.

Melatonin (MLT) is a naturally synthesized hormone secreted mainly from the pineal gland and its production is suppressed by light. It participates in the regulation of behavioral and physiological processes, including sleep and wakefulness cycle and adaptation to seasonal changes [1]. Good quality sleep contributes to better functioning, improvement in well-being, and in cognitive functions. Poor quality of sleep can be manifested by deteriorated concentration, emotional liability, impulsive behavior [2] and can also lead to a decrease in energy, anxiety, irritability, and fatigue [3]. Sleep problems affect about 20%–40% of healthy children [4]. This problem is particularly pronounced in the group of children with neurodevelopmental diseases [5] and may be related to even 80% of patients [6,7]. In pediatric practice, three groups of disorders that are most often associated with sleep disorders include ADHD, autism and mood/anxiety disorders [8]. The authors focused on the two most common disease entities, which are characterized by distorted judgment of reality in situations requiring flexibility and behavioral problems [9]. In the presented disorders, the clinical picture is heterogeneous [10] and the etiology is still unclear [11].

Autism spectrum disorder (ASD) can be defined based on the diagnostic criteria which include communication and social interaction disorders and limited repetitive patterns of behavior and interests. In addition to the fundamental symptoms, many patients also present with anxiety attacks, (self)-aggression, mood disorders, and sleep difficulties [12,13]. The first symptoms of autism are observed already in early childhood and usually persist throughout life. An increase in the incidence of ASD has been reported over recent years. According to the CDC’s Autism and Developmental Disabilities Monitoring (ADDM) report in 2000 the incidence of ASD was estimated at 1 in 150 children, in 2014 this incidence was 1 diagnosis in 59 children at the age of 8 [14]. The causes of this tendency may be different, e.g., a higher standard of medical and diagnostic care [15], higher social awareness, and the older age of parents at childbearing [16,17]. The etiology of ASD is most likely multifactorial [11], being the result of genetic and environmental factors [6]. Hallmayer et al. indicated the predominance of environmental factors, estimating the shared environmental component to be 58% and heritability to 38%. This distribution was similar among women and men [18]. So far multiple potential risk factors for ASD have been reported in the literature, including the use of illicit drugs during pregnancy, medications, organophosphate pesticides [19], fatty acid deficiency [20], gestational diabetes, and low birth weight [11]. Despite numerous scientific studies, no genetic abnormality that would account for more than 1% of ASD has been identified yet [11]. One of the hypothesis defines ASD as a collection of many forms of a rare monogenic disorder with a different etiology [21]. Ten percent of ASD can be explained as a component of a specific genetic syndrome, e.g., Rett syndrome, fragile X syndrome or tuberous sclerosis complex [22]. Pairwise ASD concordance was 31% for dizygotic twins and 88% for monozygotic twins [23]. Hallmayer et al. reported that the proband wise concordance for male twins was 77% and for female twins was 50% for monozygotic pairs whereas for dizygotic pairs it was 31% for male twins and 36% for female twins [18].

Attention deficit hyperactivity disorder (ADHD) is characterized by attention disorders, impulsive behavior and hyperactivity. Oppositional-defiant disorders affect about 25% of patients and are more prevalent among school-age patients. They are characterized by stubbornness, emotional lability and argumentative behavior. About 15%–20% of children demonstrate difficulties in adapting to general social norms, manifested by a tendency to aggressive behavior, theft, and lying; 10% of children are affected by anxiety disorders and the similar percentage of children present with depression [24]. About 65% of individuals diagnosed with ADHD in childhood show the features of the disorder in adulthood [25]. A significantly higher prevalence of comorbid lifetime psychiatric disorders is reported in adults with ADHD (77.1%) compared to control probands (45.7%) [26]. The prevalence of the disorder varies depending on the studies. The American Psychiatric Association estimates the prevalence of ADHD in the Diagnostic and Statistical Manual of Mental Disorders at 5% in the pediatric population [27]. In turn, the Centers for Disease Control and Prevention found that 9.4% of children aged 2–17 years had ever received an ADHD diagnosis [28]. There is a disparity related to the diagnosis in relation to the sex. The disorder is more prevalent among boys with the ratio ranging from 3:1 to even 9:1 [24,29,30,31]. The differences can be explained by the fact that the pattern of impulsive and hyperactive disorders is more strongly expressed in boys. The lower severity of symptoms means that girls may be undiagnosed in childhood, which explains a similar frequency in the adult population [29,32]. ADHD etiology is most likely multifactorial. The literature reports potential environmental factors such as the use of stimulants or drugs in pregnancy (nicotine and alcohol), social factors (peer and family relationships) and genetic factors. The most numerous studies suggest the effects of the dopamine pathway genes: The dopamine D4 receptor gene (DRD4), the dopamine D5 receptor gene (DRD5), and the dopamine transporter gene (DAT1) [33], but more recent publications also report the potential role of the serotonin receptor family (5-HTR) [34,35] and the serotonin transporter (5HTT) [35] in the pathogenesis of ADHD. Heritability of 0.75 to 0.91 that was robust across familial relationships (siblings, twins) was observed [36].

## 2. Melatonin in Neurodevelopmental Diseases

Normal MLT levels are crucial for the development of cognitive and behavioral functions [37]. In experimental studies MLT also shows antioxidant [38,39], anti-inflammatory [39], antidepressant [1], and neuroprotective effects [40]. The multidirectional effect of melatonin prompted researchers to analyze secretion in a group of patients with ASD and ADHD.

### 2.1. ASD

There are a number of studies (Table 1) indicating a relationship between lowered melatonin levels and the incidence of ASD.

Melatonin is a compound that crosses the placenta. Maybe low maternal MLT level may be an additional risk factor for ASD in the fetus [11,41]. Decreased MLT levels were also observed in relatives of ASD patients, which may indicate the genetic background of this phenomenon [11,37,41]. The causes of decreased MLT levels can be found in the disturbances in the metabolic pathway of MLT [42]. In 40% of patients with ASD, Pagan et al. observed increased serotonin levels with a simultaneous decrease in MLT levels occurring in 51% of ASD patients. The level of the intermediate metabolite NAS was increased, on average, in 47% of patients, which may indicate the disruption of this pathway. Their analysis showed similar disorders in first-degree relatives of patients with ASD [41]. In ASD subjects, Melke et al. demonstrated that low ASMT activity, which participates in the conversion of NAS into MLT, resulted in a decreased MLT level [37]. Abnormalities in ASMT may therefore predispose to ASD. Disorders of the circadian rhythm of MLT may correlate with the severity of ASD [43,44]. The analysis showed a 4-fold higher prevalence of ASD in boys compared to girls [19,45,46] which can be explained by a higher level of melatonin (MLT) in girls than in boys, which could translate into less frequent or less severe ASD [11]. In patients with ASD, nocturnal excretion of 6-SM negatively correlated with the severity of impairment in play and verbal communication [47]. In turn, Pagan et al. did not find a significant relationship between disorders in the levels of MLT, serotonin or N-acetyl serotonin (NAS) and the severity of autistic disorders [41]. Not all children affected by ASD manifest the disturbed circadian rhythm of melatonin secretion [48]. However, patients with decreased MLT levels complained of sleep-related problems more frequently [41].

### 2.2. ADHD

In the group of patients with ADHD, researchers report circadian melatonin arrhythmias in the form of delayed DLMO and elevated levels of excreted 6-hydroxymelatoninsulfate (6-OH MS) relative to the control group (Table 2). Van der Heijden et al. examined the differences in DLMO on the basis of MLT measurements in saliva samples in children with ADHD without sleep disorders and in children with ADHD with sleep-onset insomnia (SOI). The study did not show significant differences in sleep maintenance. However, children who reported sleep disorders presented with delayed DLMO and the delayed sleep phase [53]. Van Veen MM et al., and Bijlenga D et al. also point out the co-occurring sleep problem [54,55].

## 3. Sleep Disorders

Children affected with sleep disorders can mimic the behavior of children with ADHD [58].

### 3.1. ASD

Sleep disorders affect 50%–80% of ASD patients [19,59,60]. Many authors paid attention to the severity of ASD symptoms in children with sleep disorders [61,62,63,64,65]. Sleep disorders can present differently and take the form of difficulty falling asleep, delayed falling asleep, light sleep, frequent awakenings, shortening the sleep duration, daytime sleepiness and even insomnia [41,66,67,68]. In the Krakowiak et al. study, 529 children participated, including 303 with ASD, an analysis of parental questionnaires showed that as many as 53% of children with ASD aged 2–5 present a minimum of one sleep problem [69]. Buckley et al. observed a shortened REM sleep with an increased percentage of slow-wave sleep [66]. Wiggs et al. examined 69 children with ASD at the age of 5–16 years, the problem of insomnia affected 64% of the examined group, and the underlying was primarily behavioral problem, sleep cycle disorders and anxiety-related problems [70]. Hollway et al. found anxiety to be the strongest cause of sleep disorders in the group of ASD patients. In addition, they indicated that the quality of sleep was potentially affected by developmental regression, the age of the child and the ASD subtype—in this case sleep disorders were the most frequently reported in children with Asperger syndrome [71]. The highest percentage of sleep disorders in the group of children with ASD was presented by Liu et al. about 86%, in their work they also presented factors potentially affecting sleep disorders, such as: concomitant diseases (epilepsy, ADHD, asthma), medications taken, younger age of the child, burdened family history of sleep disorders and sleeping with parents [72]. Another problem is related to the tendency to chronic sleep disorders and a lower percentage of remission. In the study, Sivertsen et al. sleep disorders were assessed in children with ASD and in the control group when the children were 7-9 years old and 11-13 years old. The analysis showed that children with ASD were more likely to have sleep disturbances over time and were more likely to develop sleep problems relative to a healthy peer group. The estimated remission rate was 8.3% in the ASD children group compared to 52.4% in the control group [73].

### 3.2. ADHD

Depending on the study, a significant discrepancy is observed in the estimation of the prevalence of sleep disorders in ADHD patients. Meltzer et al. and Corkum et al. found coexisting sleep disorders in 25-50% of patients [8,74]. However, other studies reported much higher percentage reaching even as much as 70% [44,75]. The parents of control group children reported sleep disorders less frequently compared to the caregivers of ADHD patients [31]. Sleep disorders in ADHD children found in the subjective assessment seem to be more severe than those characterized by objective methods [31,74,75]. This is confirmed by Cohen-Zion et al. who used the objective methodology of actigraphy and polysomnography and did not observe significant differences in sleep architecture or continuity in children with ADHD compared to the control group [75]. Also, Lecendreux et al. did not show significant variability in sleep in ADHD boys compared to the control group [76]. Corkum et al. showed that total sleep time was not significantly different between children with ADHD and the control group [74]. In other studies on ADHD patients, their authors observed an increase in REM sleep percentage [77,78]. Golan et al. observed that the percentage of REM sleep accounted for 21.5% in the group of ADHD children and 18.9% in the control group [79]. In turn, Kirov et al. indicated the relationship between the higher percentage of REM sleep and daily functioning. However, the negative correlation was related to the intelligence quotient and the positive correlation was related to the severity of attention disorders in the group of ADHD patients [78]. The range of sleep disorders in ADHD children is wide and includes difficulties in falling asleep [8,31,80], sleep resistance [80], night awakenings [80] and higher activity during sleep [8,74,75]. For example, Golan et al. observed periodic limb movements in 15% of ADHD patients that did not occur in the control group [79]. Lopez et al. noted the coexistence of restless legs syndrome (RLS) in 33% of patients, a correlation was demonstrated between earlier ADHD manifestation and a more severe course, in the same study group the percentage of iron deficiency was found in 35.5% of patients [81]. Other disorders include restless sleep [8], difficulties in waking up in the morning [80] and drowziness during the day [8,75,76,79,80]. As a result of such disorders, worse quality of sleep [8,80], fragmented sleep [31] and shortened sleep are observed [8,31]. Furthermore, 50% of children with ADHD showed sleep-disordered breathing compared to 22% from the control group [79]. Complaints reported by the caregivers of ADHD patients also included hypersalivation, nocturnal enuresis, bruxism, and nightmares [75].

Treatment of ADHD includes non-pharmacological and pharmacological therapy. Drugs can be divided into two major groups, i.e., psychostimulants such as methylphenidate (MPH), which is the first-line drug, and non-psychostimulants (e.g., atomoxetine, which is a centrally acting sympathomimetic) [82]. According to some researchers, sleep disorders can be an adverse effect of stimulant drugs [24,82,83] as they can disturb the sleep pattern [84]. Schachter et al. observed that patients on MPH were affected by insomnia more frequently compared to the placebo group [85]. Galland et al. noticed the effect of MPH on sleep efficiency and observed its reduction by 6.5%, prolonged sleep onset by an average of 29 min and shortened sleep by about 1.2 h. However, the effect of MPH did not alter sleep architecture, the study compared waking time, stage 1 and 2 sleep, slow wave sleep and REM sleep [86].

Meltzer et al. indicated the negative effects of psychostimulants as they can worsen the quality of sleep and delay sleep onset [8]. Ironside et al. noted that stimulant medications caused an increase in motor activity after turning off the light in the child that tried to fall asleep. However, no changes in motor activity were observed when the child was in bed sleeping or when the child was active during the day [84]. Smits et al. did not show a significant influence of MPH on DLMO or sleep parameters. In the study group of children on MPH, insomnia had been observed before drug administration [87]. Pelham et al. did not observe an increase in sleep disorders in children who received MPH (5-15mg) that was administered in the afternoon [88].

## 4. Melatonin in the Treatment of Sleep Disorders in Children

The review presented above draws attention to the existing relationship between disorders of circadian melatonin secretion and sleep disorders in the group of patients with ASD or ADHD diagnosis. Some researchers suggest that treatment with exogenous melatonin may be effective due to potential causal effects by compensating for deficiencies. Below we present the effectiveness and safety of melatonin supplementation in sleep disorders in the group of pediatric patients. The analysis included publications meeting the following conditions: original papers, date of publication from 2000, the study involved children diagnosed with ASD or ADHD. The presented tables show only participants who took MLT and completed the study, excluding the control group.

### 4.1. ASD

Treatment should be mainly based on following the sleep hygiene and behavioral interventions. When these activities do not produce satisfactory effects, physicians should consider pharmacologic treatment. Due to lower MLT concentrations in ASD patients, studies are currently conducted on the use of MLT in the treatment of sleep disorders. Goldman et al. and Veatch et al. indicated a positive effect of MLT also in the group of patients with the normal endogenous MLT profile [48,61], which suggests that the effect of exogenous MLT was not limited to correcting deficiencies. Twenty five percent of parents of children with ASD reported that the physician suggested MLT treatment for sleep disorders in their children [89]. Cortesi et al. examined treatment effectiveness of sleep disorders using MLT, cognitive-behavioral therapy, and the combination of both methods. The best effects were obtained in patients who underwent combined therapy, then in those who received MLT alone. The lowest effectiveness was observed in patients who underwent cognitive-behavioral therapy [90].

Short-acting and sustained-release MLT is available. Despite the lack of guidelines, sustained-release MLT is recommended for children with difficulty in maintaining sleep [6,91] whereas short-acting MLT is recommended for children with difficulty in sleep onset [3]. According to Bruni et al., MLT should be administered 30 min before bedtime at a dose of 1-3 mg if it is to act as a sleep inducer. However, if it is used as a chronobiotic, it should be given 3-4 h before sleep with the initial dose of fast release MLT (0.2-0.5 mg; the maximum dose for children—3 mg and 5 mg for adolescents) [44]. In the group of patients with ASD, different melatonin dosing regiments were used, as shown in Table 3.

Owens reported MLT doses for infants (1 mg), older children (2.5–3 mg), adolescents (5 mg) and for children with special needs (0.5–10 mg) irrespective of age [91]. Due to the fact that exogenous MLT reaches its maximum concentration one hour after administration, it is usually given 30–60 min before bedtime [6,91,92]. Since the standard dose has not been established, the need for dose modification may be necessary [98]. This is due to the fact that the effectiveness of MLT is influenced by inter-individual differences in the rate of hepatic metabolism, intestinal absorption and body weight [48]. Effective dose should be established for each patient individually [104]. In turn, Malow et al. observed that the child’s age or weight was not associated with MLT dose response [59]. According to Grigg-Damberger et al., the dose should be increased by 1 mg every 2 weeks [6]. Although therapy is generally effective, even high doses of exogenous MLT did not result in the intended effect [104] in slow MLT metabolizers due to a single nucleotide polymorphism (SNP) of CYP1A2 [105].

The effects of melatonin supplementation in ASD patients are shown in Table 4. In the Wright et al. study, the sleep pattern improved by an average of 47 min [98], in the Gringras et al. study it was about 39.6 min [100]. The researchers also measured the mean prolongation in total sleep length, in the Wright et al. study by an average of 52 min [98], in the Gingras et al. study by 57.5 minutes [100].

The issue of night-time awakenings remains controversial. Garstrang et al. [94], Cortesi et al. [90], Goldman et al. [48] and Maras et al. [101] postulated that their reduction after melatonin supplementation was observed whereas Wirojanan et al. [97], Wright et al. [98] and Malow B. et al. [59] did not confirm the influence of exogenous melatonin on the change of awakening frequency. Moreover, in the literature, the researchers also indicate a mild hypnotic [82,91,106], chronobiotic effect and improved daily behavior [59,107].

Positive effects of therapy are not achieved in all children with sleep disorders. However, the obtained results seem to be satisfactory. In a study by Malow et al., all children reported improvement in sleep disorders [59]. Jan and O’Donnell showed such improvement in 82% [108] and Ayyash et al. in 78% [104]. However, Andersen et al. showed the improvement in 60% of patients [95].

### 4.2. ADHD

Sleep disturbances may exacerbate the symptoms of ADHD [30,75], especially in the morning [109]. In children with ADHD sleep restriction deteriorated from subclinical levels to the clinical range of inattention [110]. Therefore, the therapy of sleep disorders is associated with the improvement in the functioning of both children with ADHD and the whole family [75,80]. Hoebert et al. showed the effects of exogenous MLT therapy in children with ADHD and chronic sleep onset insomnia (CSOI). The mean follow-up was 3.7 years. Improvement in sleep, behavior and mood was reported in 88%, 71%, and 61% of patients, respectively [111]. Van der Heijden et al. examined 105 children with ADHD and coexisting insomnia who did not receive medications. In their study MLT supplementation was used at a dose of 3 mg or 6 mg for 4 weeks. Sleep onset advanced by 26.9 ± 47.8 min in the group on MLT and delayed by 10.5 ± 37.4 min in the placebo group (*p* < 0.0001). In the MLT group, the total time asleep also increased by 19.8 ± 61.9 min compared to the placebo group (13.6 ± 50.6 min). However, the influence of MLT on cognitive functions or problem-related behavior was not confirmed [112].

According to Danielson et al., pharmacological treatment was used in 62% of children with ADHD, 46.7% of children had received behavioral treatment whereas 23% of children had not received any treatment [28]. Based on these data, it is important to conduct research on the simultaneous use of stimulant medications and MLT. Weiss et al. obtained positive effects of combination therapy (sleep hygiene with MLT at a dose of 5 mg for 30 days) in children with ADHD who used stimulant medications. Insomnia was initially decreased by 16 min compared to the placebo control group [113]. TjonPian Gi et al. reported that the time of falling asleep after MLT administration varied between 15–64 min [114] in ADHD subjects on MPH and MLT at a dose of 3 mg. In turn, Masi et al. reported improved sleep in 60.8% of ADHD patients treated with MPH after MLT therapy (mean dosage 1.85 ± 0.84 mg/d) [82]. Schemes of conducted research and effects of melatonin therapy in the group of patients with ADHD are summarized in Table 5 and Table 6.

## 5. Safety of Exogenous Melatonin in Pediatric Patients

Melatonin treatment is widely regarded as safe [19,59,108] and adverse effects are rarely reported in the literature, which is the additional advantage of MLT due to the fact that there are reports on more prevalent side effects in the groups of ASD patients who received psychotropic medications compared to their peers without ASD [13]. Andersen et al. reported mild side effects, i.e., increased nocturnal enuresis, morning drowsiness. One child had worse sleep after MLT administration [95]. Other symptoms include headache and dizziness, gastric complaints (e.g., loose stools) [59,117]. While Owens paid attention to the possibility of premature puberty by the suppressing the pituitary-gonadal axis [91], Van Geijlswijk et al. did not confirm deviated pubertal development in the group of subjects on MLT (mean period of administration was 3.1 years) with the mean dose of 2.69 mg compared to the age and sex-matched control group [118]. Malow et al. noted the severity of fatigue during the dose increase, which resolved after repeated reduction of supplementation [103]. Babineau et al. in their study indicated the safety and effectiveness of short-term MLT therapy in children [119].

Long-term studies on melatonin supplementation were conducted by Malow et al. for 104 weeks [103], Hoebert et al. for a diameter of 3.66 ± 0.12 years [111], Van Geijlswijk et al. for one to 4.6 years [118], Carr et al. for an average of 4.3 years [120], and Jan and O’Donnell for a period of several weeks up to 4 years [108]. Safety of MLT in children was confirmed by these studies. In the research by Hoebert et al. [111] and TjonPian Gi et al. [114] it was noted that discontinuation of exogenous MLT was associated with recurrent sleep disorders and further supplementation was required. In the study of Malow et al. intermittently long-term therapy resulted in a deterioration in sleep, but the patients’ condition remained better than it was at baseline [103]. Carr et al. compared MLT doses in 33 patients without previous exogenous MLT therapy in whom the treatment lasted from 9 months to 3.8 years (mean 2.5 years) with the doses in 11 patients previously on MLT from 5 months to 9.6 years (mean 5.1 years). The final dosage of exogenous MLT was not statistically significantly different in both subgroups, suggesting that tolerance in MLT patients did not develop over long-term use [120]. The same conclusions were presented by Jan and O’Donnell [108]. The effects of MLT supplementation after the first weeks were comparable with those observed after 3 months [114]. Masi et al. and Weiss et al. found that combination therapy of MPH and MLT was effective and safe [82,113].

## 6. Discussion

This review indicates the significance of sleep disorders in the pediatric group, especially in children with neurodevelopmental disorders.

It remains unclear whether sleep problems are part of the clinical picture of ASD and ADHD or are provoked by anxiety or behavioral disorders. The authors point out circadian arrhythmias in this group of patients. However, it is certain that poor sleep quality translates into poorer functioning, not only for the patient but also for their carers. It should be remembered that the first action of a clinician should be to look for the cause of the problem and convince the patient and his family to observe proper sleep hygiene and only then pharmacotherapy.

Recommendations for the treatment of sleep problems in children with neurodevelopmental disorders involves behavioral interventions aimed at maintaining the patient’s activity during the day, together with the reduction and control of stimuli during the night period. The melatonin treatment and adherence to sleep hygiene, collectively with planning regular bedtime and waking up time, and avoiding daytime naps are important. Melatonin is one of the most commonly used agents for sleep disorders in the group of pediatric patients [98].

The presented data testify to the beneficial effect of melatonin supplementation. The authors seem to agree that exogenous melatonin improves sleep delay as well as extends overall sleep duration in the group of patients with ASD and ADHD. British Society of Psychopharmacology in 2018 [121] based on Rossignol and Frye data from 2011 (evidence level Ia) [107] and Cortesi et al. from 2012 (evidence level Ib) [90] recommends the use of melatonin in children with ASD. Data on the effects of melatonin on behavior are less numerous. In the group of ASD patients, researchers agree that melatonin has a positive effect on behavior. Van der Heijden et al. and Mohammadi et al. they did not find a beneficial effect of melatonin supplementation on the behavior of patients with ADHD [112,115], interestingly, Hoebert et al. described an improvement in patient behavior [111]. These differences may be related to the duration of therapy because Hoebert et al., conducted the longest observation of patients—an average of 3.66 ± 0.12 year [111]. The obtained sleep improvement in the patient also improves the quality of life of the caregiver and the whole family of the patient.

Of note, despite the widespread use, MLT is not a medication but an over-the-counter dietary supplement. As a result, it is not subject to such strict monitoring of adverse effects compared to prescription drugs. In the above studies, adverse reactions were absent or rare. MLT is considered safe and well-tolerated. High availability and a low price of MLT are its advantages. The biggest benefit of melatonin is its high effectiveness. Discontinuation of MLT treatment is often associated with recurrent sleep disorders.

The review presented extensively presents the protocols used for melatonin supplementation, which may facilitate the work of clinicians because there has not been a uniform dosage standard to date.

## 7. Limitations

The review presented extensively presents the protocols used for melatonic supplementation, which may facilitate the work of clinicians because there has not been a uniform dosage standard to date.

The review has several limitations:-This is not a systematic review carried out in accordance with the PRISMA guidelines;-research groups are not limited to patients with ASD and ADHD;-no homogeneous test report;-the use of various melatonin preparations in studies;-depending on the publication, various parameters (assessment of sleep disorders, assessment of behavior) were assessed using various methods (objective - actigraphy, polysomnography and subjective - sleep diaries, questionnaires); and-lack of careful monitoring also of light side effects.

## 8. Conclusions

Despite the limitations, our review confirms the efficacy and safety of melatonin in the treatment of sleep disorders in the group of pediatric patients. We noticed a small amount of work on the effect of melatonin on behavior and the need to design a large long-term study using both subjective and objective methods to assess the effectiveness of melatonin.

## Figures and Tables

**Table 1 brainsci-10-00219-t001:** Melatonin level assessment in the group of patients with autism spectrum disorder (ASD).

References	Number of Patients	Material	Results
Nir I et al., 1995 [49]	10 patients with ASD 5 controls	serum	1. Similar results of mean daily melatonin levels were measured in serum in the group of patients with ASD and the control group. 2. Patients with ASD showed lower melatonin levels at night and higher during the day relative to the control group.
Tordjman S et al., 2005 [47]	49 patients with ASD 88 controls	urine	1. Low melatonin levels were observed in 63% (31/49) of patients with ASD. 2. In pre-pubertal period, a reduction in 6-SM was apparently observed in patients with ASD relative to the control group.
Melke J et al., 2008 [37]	43 patients with ASD 34 parents 48 controls	plasma	1. Low melatonin levels were observed in 65% of patients with ASD. 2. There is a genetic potential for reduced melatonin levels in ASD, as asymptomatic parents also have abnormally low melatonin levels. 3. The results suggest that in patients with ASD and low melatonin levels we can get sleep improvement through supplementation.
Tordjman S et al., 2012 [43]	43 patients with ASD 26 controls	urine	Patients with ASD showed lower levels of nocturnal 6-SM excretion in urine than those in the control group.
Pagan C et al., 2014 [41]	278 patients with ASD 129 unaffected siblings 377 parents 416 controls	plasma	Melatonin levels in patients with ASD and their relatives were significantly lower than in the control group.
Abdulamir HA et al., 2016 [50]	60 patients with ASD 26 controls	serum	1. Patients with ASD presented lower levels of melatonin than the control group. 2. A relationship has been demonstrated between the severity of autism and lower melatonin levels.
Benabou M et al., 2017 [51]	157 patients with ASD 100 unaffected siblings 264 parents	plasma	Phenotypic variation and changes in melatonin levels in patients with ASD and their families are the result of environmental and genetic factors.
Braam W et al., 2018 [11]	60 mothers of a child with ASD 15 controls	urine	1. Mothers of children with ASD had significantly lower levels of 6-SM in urine compared to the control group. 2. Mothers of children with ASD were older at the time of child birth compared to mothers of children without ASD.
Maruani A et al., 2019 [52]	78 patients with ASD 90 unaffected relatives 47 controls	plasma	1. Patients with ASD presented lower levels of melatonin than their relatives and people from the control group. 2. Lower levels of melatonin were observed in relatives of patients with ASD, however, the result was not statistically significant.

**Table 2 brainsci-10-00219-t002:** Melatonin level assessment in the group of patients with attention-deficit/hyperactivity disorder (ADHD).

References	Number of Patients	Material	Results
Van der Heijden KB et al., 2005 [53]	87 patients with ADHD-SOI 33 patients with ADHD-noSOI	saliva	ADHD patients with concomitant chronic idiopathic insomnia at the onset of sleep presented significantly delayed DLMO and sleep phase relative to the ADHD-noSOI group.
Van Veen MM et al., 2010 [54]	34 patients with ADHD 38 controls	saliva	1. Melatonin production in the ADHD group began 83 min later than the control group. 2. Patients with ADHD showed less effective sleep and longer sleep delay.
Baird AL et al., 2012 [56]	13 patients with ADHD 19 controls	saliva	1. In the group of patients with ADHD disturbed rhythm of melatonin relative to the control group was observed.
Bijlenga D et al., 2013 [55]	12 patients with ADHD 12 controls	saliva	1. In the group of people with ADHD, DLMO was delayed by about 1.5 h relative to the control group. 2. An average one hour longer interval between the onset of DLMO and onset of sleep was observed in subjects with ADHD relative to the control group.
Büber A et al., 2016 [57]	27 patients with ADHD 28 controls	urine	Patients with ADHD had significantly higher levels of total 24-h urinary excretion of 6-OH MS than controls.

(ADHD-SOI)—ADHD-related sleep-onset insomnia, (ADHD-noSOI)—ADHD without sleep-onset insomnia.

**Table 3 brainsci-10-00219-t003:** Melatonin supplementation in ASD patients.

References	Study Design	Number of Patients	Age	Diagnosis	Melatonin Dosage	Time of Melatonin Supplementation (before Bedtime)	Duration of Melatonin Supplementation
Gupta R, et al. 2005 [92]	Retrospective study	9	2–11 years	ASD	2.5–5 mg	45 min	
Giannotti F, et al. 2006 [93]	Open label	20	2.6–9.1	ASD	3 mg (1 mg FR and 2 mg CR)–6mg	30–40 min	2 years
Garstrang J, et al. 2006 [94]	DB-RCT	7	4–16	ASD ADHD	5 mg		4 weeks
Andersen IM, et al. 2008 [95]	Retrospective study	107	2−18	ASD	0.75–6 mg	30–60 min	1.8 ± 1.4 years
Wasdell MB, et al. 2008 [96]	DB-RCT	50	2.05−17.81	Severe intellectual loss Cerebral palsy Epilepsy Visual impairment Lack of mobility ASD	5 mg CR	20–30 min	10 days
Wirojanan J, et al. 2009 [97]	DB-RCT	12	2–15.25	ASD Fragile X syndrome	3 mg	30 min	2 weeks
Wright B, et al. 2011 [98]	DB-RCT	16	4–16	ASD	2–10 mg	30–40 min	3 months
Cortesi F, et al. 2012 [90]	DB-RCT	74	4–10	ASD	3 mg CR	21:00	12 weeks
Gringras P, et al. 2012 [99]	DB-RCT	51	3.7–15	DD alone DD and epilepsy DD and ASD DD, ASD, epilepsy DD and “other”	0.5–12 mg	45 min	12 weeks
Malow B, et al.2012 [59]	Open label	24	3−10	ASD	1–6 mg	30 min	14 weeks
Goldman S, et al. 2014 [48]	Open label	9	3−8	ASD	1–3 mg	30 min	3–6 weeks
Gringras P, et al. 2017 [100]	DB-RCT	58	2−17	ASD SMS	2–5 mg of PedPRM		13 weeks
Maras A, et al. 2018 [101]	DB-RCT	51	2−17.5	ASD Neurogenetic disorders	2–10 mg of PedPRM	30−60 min	52 weeks
Schroder CM, et al. 2019 [102]	DB-RCT	58	2−17.5	ASD SMS	2–5 mg of PedPRM	30−60 min	13 weeks
Malow BA, et al. 2020 [103]	DB-RCT	74	2−17.5	ASD SMS	2–10 mg of PedPRM	30−60 min	104 weeks

DB-RCT—double blind, randomized control trial, CR—controlled-release, FR—fast release, DD—developmental delay, PedPRM—pediatric-appropriate 3 mm diameter prolonged release melatonin minitablet, SMS—Smith–Magenis syndrome.

**Table 4 brainsci-10-00219-t004:** Effects of melatonin supplementation in ASD patients.

References	Sleep Latency	Total Duration of Sleep	Behavior	Night−Wakings
Gupta R, et al. 2005 [92]	+	+		
Garstrang J, et al. 2006 [94]	+	+	+	+
Wasdell MB, et al. 2008 [96]	+	+		
Wirojanan J, et al. 2009 [97]	+	+		−
Wright B, et al. 2011 [98]	+	+		−
Gringras P, et al. 2012 [99]	+	+	+	
Cortesi F, et al. 2012 [90]	+	+		+
Malow B, et al.2012 [59]	+	−	+	−
Goldman S, et al. 2014 [48]	+			+
Gringras P, et al. 2017 [100]	+	+		
Maras A, et al. 2018 [101]	+	+		+
Schroder CM, et al.2019 [102]			+	

(+) improvement (**−**) without any significant difference.

**Table 5 brainsci-10-00219-t005:** Melatonin supplementation in patients with ADHD.

References	Study Design	Number of Patients	Age	Diagnosis	Melatonin Dosage	Time of Melatonin Supplementation (before Bedtime)	Duration of Melatonin Supplementation
Weiss MD, et al. 2006 [113]	**DB-RCT**	19	6.5–14.7 years	ADHD	5 mg	20 min	10 days
Van der Heijden KB, et al. 2007 [112]	**DB-RCT**	53	6–12	ADHD	3–6 mg FR	19:00	4 weeks
Hoebert M, et al. 2009 [111]	**DB-RCT**	94	12.39 ± 0.25	ADHD	3–6 mg	18:30–23:00	3.66 ± 0.12 year
Mohammadi MR, et al. 2012 [115]	**DB-RCT**	26	7–12	ADHD	3–6 mg		8 weeks
Mostafavi SA, et al. 2012 [116]	**DB-RCT**	26	7–12	ADHD	3–6 mg		8 weeks
Ayyash HF, et al. 2015 [104]	**Prospective, observational, naturalistic study**	45	6.3 ± 1.7	Intellectual disability ASD ADHD	2.5–10 mg	30 min	326 days
Masi G, et al.2019 [82]	**Observational, naturalistic study**	74	11.6 ± 2.2	ADHD	1–5 mg	1–2 h	4 weeks–12 months

DB-RCT—double blind, randomized control trial, FR—fast release.

**Table 6 brainsci-10-00219-t006:** -Effects of melatonin supplementation in patients with ADHD.

References	Sleep Latency	Total Duration of Sleep	Behavior	Cognition	Quality of Life	Frequent Awakenings
Van der Heijden KB, et al. 2007 [112]		+	−	−	−	
Hoebert M, et al. 2009 [111]	+		+			
Mostafavi SA, et al. 2012 [116]	+	+				
Mohammadi MR, et al. 2012 [115]	+	+	−			
Ayyash HF, et al. 2015 [104]	+	+				+

(+)—improvement (−)—without any significant differences.

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
