# Peer review of "Efficacy and Safety of Melatonin Treatment in Children with Autism Spectrum Disorder and Attention-Deficit/Hyperactivity Disorder—A Review of the Literature"

_brainsci, 2020, doi:10.3390/brainsci10040219_

Round 1
Reviewer 1 Report
The authors sent a review on melatonin efficacy and safety in neurodevelopmental diseases, focused on autism spectrum disorder (ASD) and attention-deficit/hyperactivity disorder (ADHD). Generally, a systematic review is a review of a clearly formulated question that uses systematic and reproducible methods to identify, select and critically appraise all relevant research, and to collect and analyze data from the studies that are included in the review. According to my opinion, there is not clear how many articles were enrolled in the study, which database was used, and a way for accepting or omitting research articles. Therefore, “A systematic review” should be dropped from the title.
A review includes a basic information on above mentioned developmental disorders, their melatonin metabolism changes, sleep disorders comorbidities, and treatment possibilities focused mainly on melatonin supplementation.
Questions and remarks:
Line 60-64: a reference should be added
Line 107: “The difference between the sexes decreases with age”. Why?
Line 112-113: The reference is 23 years old. Is there any new research on heritability topic in ADHD?
Line 128 and below: A bifurcation of the sleep wake cycle due to melatonin disruption should be mentioned (Cortese et al. Sleep Med 2010)
Line 167-168: “…chronic sleep disorders and lower percentage of remission” – it is not clear, what do you mean
Line 196: Besides periodic limb movements restless leg syndrome and ferritin deficiency should be mentioned with adequate references
Line 212-213…. “the effect of MPH did not alter sleep architecture”. Do you mean cycles of NREM-REM sleep and the amount of NREM and REM sleep?
Line 215: Change Iron side to Ironside
Line 329-332: The sentences should be formulated better, more clearly
Author Response
In response to Reviewer 1's comments, the following changes have been made:
The authors modified the title of the manuscript, removing the "Systematic review".
Line 89-91 - the authors explained what the potential for equalisation of the prevalence of diseases in adulthood is
Lines 94-98 - data related to potential genetic factors have been updated
Lines 135 - 150 - paragraph on the relationship between melatonin secretion in patients with ASD and ADHD was developed
Lines 177-181 - information on the frequency of remissions and new sleep disorders over the years has been supplemented
Lines 205-208 - information on RLS and ferritin deficiency has been supplemented
Lines 224-225 - it has been specified what data was taken into account during the sleep architecture assessment details of what data were taken into account when evaluating sleep architecture
Line 227 - Ironside refund has been corrected
Lines 345-353 - data on long term observations on melatonin supplementation have been organized
Reviewer 2 Report
BRAINSCI-726876
Title: Efficacy and safety of melatonin treatment in children with autism spectrum disorder and attention-deficit/hyperactivity disorder. Systematic review.
The manuscript is a review of the literature on the role of melatonin in ADHD and ASD and role of melatonin treatment on sleep disorders.
Some point must be improved:
Major point:
- The introduction sessions of both ASD and ADHD are overly general. They should be streamlined to facilitate the reading.
- The first section of the introduction on ASD and on ADHDneeds to be improved to better understand the link between each other. Sometimes ideas are not clearly presented and it becomes difficult to follow further discussion.
- It would be appropriate to specify the aims and methods of the study.
- Limitations in the organization and methodology curtail the paper potential impact:
- In the methodology, a quantification of the number of works analyzed should be indicated.
- In the methodology, the process that led to the organization of this review work should be clarified.
- Critical details are omitted such as key words, search portals, and the “time stamped” results of these searches.
- The literature on sleep disorders in ASDs is not exhaustive. A wider overview would be helpful.
- The link between ASD and ADHD with melatonin must be supported.
- Minor point:
in line 125 there is an underlined reference.
Author Response
The following changes have been introduced in response to Reviewer 2's comments:
Authors modified the title of the manuscript, removing the "Systematic review".
The introduction has been modified, removing a significant part of the information that was not directly related to the presented topic, attention was paid to the common features of the described disorders.
The data on melatonin secretion in patients with ASD and ADHD and the paragraph regarding sleep disorders in patients with ASD have been supplemented.
Attention was paid to how melatonin secretion disorders contributed to attempts to treat sleep disorders by supplementation. Link underline was removed.
Reviewer 3 Report
Thank you for inviting me to review the paper entitled “Efficacy and safety of melatonin treatment in children with autism spectrum disorder and attention-deficit/hyperactivity disorder. Systematic review.” The review is aimed at presenting the role of melatonin in two neurodevelopmental disorder, ASD and ADHD, and at discussing the influence of melatonin treatment on sleep disorders in these populations.
I have carefully read the paper and, despite this is presented as a systematic review, it does not follow the methodology of a systematic review.
- First, the paper should be divided in the traditional sections of a manuscript, i.e. Introduction, Methods, Results, and Discussion. In the Introduction, the authors should present the information reported in sections 1 to 6. The authors can choose to divide information in separate paragraphs. Nevertheless, in my opinion, the information reported in the introduction are too detailed and should be summarized. On the other hand, the authors should focus more on the results.
- Importantly, a rationale for the realization of the review should be explicitly stated at the end of the introductory part. In fact, it is not clear what this paper adds to the current literature, since many systematic reviews and meta-analyses have been recently published on the topic (see for instance Abramov et al. 2019, Jenabi et al. 2019, Abdelgadir et al. 2018, Gringras et al 2017, Xie et al 2017, Anand et al 2017, etc.)
- One major caveat is represented by the absence of methodology. Was the review conducted according to PRISMA guidelines? If so, which were the search terms? In which databases did the authors search? How many papers did they retrieve, how many were excluded and how many were included? Which inclusion criteria were chosen (Participants, Intervention, Comparison, Outcome, Study Design)? A PRISMA flow chart would be also desirable. I strongly suggest to consult the Prisma statement (http://www.prisma-statement.org/), or at least to read other systematic reviews to better understand the methodology usually followed.
- As far as concern the results, they should be divided according to the two focus selected by the authors, efficacy and safety (if I have understood correctly)
- I have appreciated the presentation of the studies in tables, and in particular the tables showing the effect on specific outcomes. However, they should be accompanied with specific paragraphs with a detailed description of study characteristics in the main text (e.g. number and age of participants, study design, study duration, melatonin dosage, outcome scales, etc.).
- Finally, the authors should critically discuss the findings and the limitation of the review, in light of the included studies, and give advice about implications for clinical practice and research-
If the authors do not decide to follow the rules of a systematic review, they should firstly change the title of the paper, shorten the introduction (especially when presenting the two conditions, e.g. it is not necessary to thoroughly describe the evolving epidemiology of ASD…) and add at least some critical comments after the presentation of the findings.
In conclusion, this paper is not ready for publication. A radical change in methodology and/or conceptualization should be taken into account by the authors.
Author Response
In response to Reviewer 3's comments, the following changes have been made:
The authors modified the title of the manuscript, removing the "Systematic review".
The introduction has been modified, removing much of the information that is not directly related to the presented topic, attention was paid to the common features of the described disorders.
The manuscript presented summarizes the abnormalities of melatonin secretion in the study group as well as the effect and safety of melatonin treatment in sleep disorders.
Referring to other reviews:
- Abramov et al. 2019 do not refer to melatonin in patients with ASD and ADHD
- Compared to Jenabi et al. 2019, there are fundamental differences:
· the authors focus on ASD
· the authors compare 10 publications, in our publication there are 14 of them
· we do not agree with the maximum dose of melatonin used in the publication Gringras 2017
· the authors present the size of the study group, we focus on the size of the group covered by the supplement excluding the control group and people who have not completed the study, because there is no separation between the control group and those taking melatonin, and the dose of melatonin is given can be misinterpreted the size of the study
· the authors present the total time of the study, we focus only on the period during which the patients actually supplemented melatonin.
- Compared to Abdelgadir et al. publication .2018 there are the following differences
· both reviews concern both ASD and ADHD
· the authors compare a total of 7 publications from the above mentioned fields, in our publication there are 21 in total
· the authors present the size of the study group, we focus on the size of the group covered by the supplement excluding the control group and people who have not completed the study, because there is no separation between the control group and those taking melatonin, and the dose of melatonin is given can be misinterpreted the size of the study
· he lack of application of the time intervals does not provide information on the safety of long-term therapy
- Compared to Gringras and others. 2017, there are the following differences:
· in their work, the authors focus on the research conducted and do not make a broad comparison with other publications
· the above mentioned publication was compared in our manuscript with other publications
- Compared to Xie et al. 2017 are as follows
· the above mentioned article does not discuss the use of melatonin in asd and adhd patients
- Compared to Anand et al. 2017, there are the following differences
· the authors focus on ASD
· the authors do not give any indication of the potential effects of melatonin on the publications in question.
The manuscript has been completed with the restrictions that occurred during writing.
Round 2
Reviewer 1 Report
The authors sent a revised version of their article on melatonin treatment in development disorders, particularly autism spectrum disorder and ADHD. They accepted almost all remarks of reviewers and answered sufficiently their questions. The title of the manuscript was corrected, and arrangements of the manuscript was changed. The introduction was shortened, the other parts of the manuscript were corrected according to reviewers´ opinion, Tables 1-4 were modified, tables 5-6 were added, and the references were completed.
Author Response
The authors would like to thank the reviewer.
Reviewer 3 Report
I appreciate the authors' efforts to improve the manuscript, but I still have some concerns.
1. Introduction: Please explain, at leats the first time, that N-acetyl-5-methoxytryptamine is the technical noun for melatonin., i.e. “Melatonin, also called N-acetyl-5-methoxytryptamine, is the main regulator of circadian rhythm and was first isolated in 1958.”
2. Melatonin for ASD: Please note that recently the guidelines of the British Society of Psychopharmacolgoy (see doi: 10.1177/0269881117741766) have suggested the use of melatonin for ASD, with evidence level Ia. This is an important issue that should be introduced by the authors, which conversely state that there are no guidelines supporting the use of melatonin in ASD.
3. In Table 3 and 4 the authors should specify the design of the included study (e.g. RCT double-blind, observational parallel without randomization, open-label, etc.) to better understand the validity of the results obtained
4. I still don’t see a critical discussion of the results obtained by the authors. In summary, what are the beneficial effects of melatonin and why could it be important to use in these groups of people? (e.g. better sleep à less behavioral problems à less burden for caregivers, etc.) What should clinicians do? What do we need for future research? For instance, I see that there are no studies about the efficacy of melatonin in adults with ASD or ADHD and this should be a critical issue to discuss (see doi: 10.1016/j.psychres.2019.04.013.). Or maybe the authors choose to select only study including children? In such case, this choice should be clearly stated and justified.
5. Finally, before moving to conclusions, the authors should add a paragraph about the limitations of the study, in particular the fact that it is not a systematic review conducted according to PRISMA guidelines.
Author Response
In response to the Reviewer's comments, the following changes were introduced:
- Taken out of N-acetyl-5-methoxytryptamine is the technical noun of melatonin
- As suggested by the reviewer, the recommendations of the British Society of Psychopharmacology were added, which are based on the work of Rossignol et al (2011), Gringras et al (2012) and Cortesi et al (2012).The manuscript presented a publication of Cortesi from 2012. The Rossignol publication is a systematic review and meta-analysis in which 5 publications were used, 4 of them were presented in the manuscript, the publication of Mc Arthur et al was omitted because we focused on publications from the 21st century. A 2012 Gringras publication has been added to the manuscript.
- Tables 3 and 5 have been supplemented with a study design
- The discussion was modified, the authors supplemented it with infermotion recommended by the Reviewer. Highlighted in the paragraph about melatonin treatment that this manuscript is limited to the pediatric population. Attention is paid to a small amount of research on the effects of melatonin on children's behavior. The need to use both objective and sub-subjective methods to assess the effectiveness of melatonin was underlined.
- The paragraph regarding limitations of the presented manuscript has been supplemented.